# Non-Parametric State-Space Models Over Datapoints and Sequence Alignments

## Abstract

Non-parametric models are flexible and can leverage a context set to express rich mappings from inputs to outputs. However, these methods often scale super-linearly in context size, e.g., attention-based methods scale quadratically in the number of data points, which in turn limits model expressivity. In this work, we leverage advances in state-space modeling and introduce Non-Parametric State Space Models (NPSSM). We find that NPSSMs attain similar performance to existing non-parametric attention-based models while scaling linearly in the number of datapoints. We apply NPSSMs to the task of genotype imputation, where the linear scaling enables larger context sets resulting in competitive performance relative to other methods and widely used industry-standard tools. We also demonstrate the effectiveness of NPSSMs in the context of meta-learning where the ability to efficiently scale to larger training sets provides more favorable compute-to-accuracy tradeoffs.

## 1 Introduction

Machine learning (ML) models often benefit from having access external datasets at inference time. In meta-learning, we often seek to condition the model on a previously unseen training set. In biology, data can be aligned to known sequences which can be used to improve predictions. More generally, non-parametric ML uses the training set at test time and includes both classical algorithms (e.g., k-nearest neighbors, support vector machines), as well as modern methods such as retrieval-augmented language models and non-parametric transformers.

However, non-parametric ML algorithms can often have high computal and memory overhead. For example, while modern transformer-based methods can be used to attend to representations of sequences or datapoints, their computational complexity grows quadratically in the size of their context. This mirrors classical kernel methods that also have quadratic complexity. Addressing these computational requirements typically requires approximations that compress the context size, such as inducing point methods.

In sequence modeling, the quadratic computational complexity of transformers has motivated the development of alternative architecture such as state-space models (SSMs). They have received significant attention because of their ability to capture long context without the quadratic compute cost of attention-based architectures.

This paper applies state-space models to non-parametric ML and leverages their long context to model interactions between large sets of datapoints. Specifically, we introduce NP-SSMs, a supervised learning architecture that takes as input a collection of labeled and unlabeled datapoints and produces accurate predictions by querying the labeled data. Key to this approach are novel non-parametric layer blocks that compute representations over datapoints using bidirectional SSMs. These blocks are analogous to existing architectures that apply attention to datapoints or alignments.

We use the NP-SSM architecture to parameterize a probabilistic model $p(x \| D)$ of data x conditioned on a dataset D, which we call the SSM neural process. Like other types neural processes (NPs), these can be trained using forms of maximum likelihood, as well as forms of meta-learning where datasets are sampled from a meta-training set. Leveraging SSMs enables these NPs to handle much larger context sizes, which in turn can significantly improve their performance what more data is available.

We apply our novel models and architectures across benchmarks in the neural process literature, where we observe significant reductions in memory usage, as well as problems in biology, where they outperform classical transformer-based approaches. Overall, we find that NP-SSMs can serve as a drop-in replacement for many applications that are currently the domain of non-parametric transformers or neural processes.

Our main application area is genotype imputation in statistical genetics, where inputs consist of aligned sequences of genetic variants and our goal is to fill in missing positions obtained using an inexpensive assay to match the performance of an accurate sequencing instrument. We find that our method outperforms highly engineered state-of-the-art software packages widely used within commercial genomics pipelines (Browning et al., 2018), indicating that our technique has the potential to impact real-world systems.

In summary, our contributions are:

- We develop an architecture for non-parametric machine learning called the non-parametric state-space model, made possible by SSM blocks applied to tokens that represent datapoints.

- We combine this architecture with training objectives that maximize expected log-likelihood across datasets, and that are naturally suited for meta-learning.

- We apply our model to genotype imputation, an important problem in statistical genetics, and we show that by leveraging large sets of alignments, our approach obtains state-of-the-art performance and outperforms specialized software packages.

## 2 BACKGROUND

### 2.1 NON-PARAMETRIC GENERATIVE MODELS

Most supervised models are parametric: given a dataset $D_{\text{train}} = \{x^{(i)}, y^{(i)}\}_{i=1}^n$, with input features $x \in \mathcal{X}$ and labels $y \in \mathcal{Y}$, the goal is to learn a set of parameters $\theta \in \Theta$ that yield an accurate predictive model $f(x; \theta)$. However, modern parametric models can have up to hundreds of billions of parameters, making training and prediction computationally expensive. Non-parametric or semi-parametric models of the form $y = f(x \| D_{\text{train}}; \theta)$ leverage the training dataset $D_{\text{train}}$ at inference time to reduce costs by retrieving information from $D_{\text{train}}$ instead of having to compress $D_{\text{train}}$ into parameters $\theta$. Multiple deep learning architectures fall under the non-parametric/semi-parametric archetype, including memory-augmented architectures (Graves et al., 2014; Santoro et al., 2016), retrieval based language models (Grave et al., 2016; Guu et al., 2020; Rae et al., 2022; Min et al., 2023), models utilizing Retrieval Augmented Generation (RAG) (Lewis et al., 2020), and non-parametric transformers (Kossen et al., 2021; Rao et al., 2021; Notin et al., 2023b).

However, the flexibility of non-parametric models comes with a different computational cost, as most models either scale superlinearly in the size of the context or require heuristics to make handling the context set tractable.

### 2.2 META-LEARNING

Meta-learning is a natural application for non-parametric models. We are given a meta-dataset $\mathcal{D} = \{\mathcal{D}^{(i)}\}_{i=1}^D$, where each element $\mathcal{D}^{(i)}$ consists of a context set $\mathcal{D}_c^{(i)} = \{(x_c^i, y_c^i)\}_{i=1}^n$ and query set $\mathcal{D}_q^{(i)} = \{(x_{\text{query}}^i, y_{\text{query}}^i)\}_{i=1}^m$. The goal of meta-learning is to fit a function $y_{\text{query}} = f(x_{\text{query}}; \mathcal{D}_c)$ that'll work for an arbitrary pair of context and query datasets. In meta learning, the meta dataset will be composed of distinct tasks that have a similar structure, e.g. fitting polynomials, tasks over Multiple Sequence Alignments (MSAs), or some other structured context and the end goal is to be able to generalize to context sets that haven't been seen that have a similar structure to those seen during meta training.

### 2.3 STATE-SPACE MODELS

Most existing deep non-parametric architectures use utilize some form of attention (Vaswani et al., 2017), causing those architectures to scale quadratically in the length of the context set. In practice,

this quadratic scaling often limits the usability of these architectures and necessitates some sort of approximate/sparse attention or some other heuristic to mitigate this scaling (Zaheer et al., 2021; Wang et al., 2020; Jaegle et al., 2021; Rastogi et al., 2023) .

State Space Models (SSM)s are a class of sequence models that have recently begun gaining popularity due to their performance on sequence modeling and their favorable performance characteristics (Gu et al., 2020; 2021; 2022; Smith et al., 2022; Dao et al., 2022; Schiff et al., 2024). In particular, the Mamba model is a selective SSM that has been proposed as a drop in replacement for self-attention that maintains most of the performance of traditional self attention for most tasks, but with linear scaling in the length of the sequence (Gu & Dao, 2024).

## 2.4 GENOTYPE IMPUTATION

Our specific motivating example is genotype imputation. Human genetic sequences can be thought of as a sequence $y \in \{A, C, T, G\}^n$. While there exist methods to read the entire sequence $y$, these methods can be costly. Instead, we can consider reading some smaller sub-sequence $x \in \{A, C, T, G\}^t$ where $t << n$. If we leverage some statistical properties of the genome, we can construct $x$ so that we can use it to reconstruct the full sequence $y$ while only reading $t$ positions. This problem of reconstructing the full $y$ given some subset $x$ is called genotype imputation. We use a relatively cheap device called a microarray to obtain a set $x$ for some individual, and then using some statistical methods we reconstruct the full $y$ given $x$ and some context set $H_{\text{ref}} = \{(x^{(i)}, y^{(i)})\}_{i=1}^k$ that contains the full sequences for $k$ individuals (Li et al., 2009). Existing methods for genotype imputation are all based on an HMM from Li & Stephens (2003), they leverage the fact that due to recombination, the full sequence $y$ can be reconstructed as a mosaic of the $y^{(i)}$s in $H_{\text{ref}}$. Therefore the problem of imputing $y$ becomes finding the most likely path through $H_{\text{ref}}$ given the observed sub sequence $x$. Given how $y$ is represented as a mix between the $k$ sequences in $H_{\text{ref}}$, increasing the number of individuals $k$ in $H_{\text{ref}}$ tends to subsequently increase performance. Existing methods like Beagle and Impute (Browning et al., 2018; Rubinacci et al., 2020) scale quadratically in $k$, limiting performance and necessitating workarounds.

Non-parametric approaches provide a compelling case for use in genotype imputation. They admit a meta learning approach that fits some function of the form $y = f(x; H_{\text{ref}})$ that can be trained on multiple different $(x, y, H_{\text{ref}})$ tuples. By included $(x, y)$ pairs trained on different microarrays with different typed positions, and with different compositions of $H_{\text{ref}}$, we can define a model that can be applied to any arbitrary region of the genome for any microarray. However, existing non-parametric models often utilize an attention operation that is also quadratic in $k$, running into the same bottleneck that the existing HMM methods encounter. In this work, we propose to utilize State Space Models in lieu of attention operations, giving us the capability to scale *linearly* in $k$ while retaining performance, allowing for increased performance due to the ability to scale to larger values of $k$.

## 3 NON-PARAMETRIC STATE SPACE MODELS

This work applies advances in long-context state space models to non-parametric machine learning. Specifically, we propose an architecture and associated training objectives that take as input a large dataset and output a dataset representation that is useful for tasks such as prediction and imputation.

**Notation and Inputs**   Given a training set $D_{\text{train}} = \{x^{(i)}, y^{(i)}\}_{i=1}^n$, we seek to make an accurate prediction at a new datapoint $x_{\text{query}}$ using a model $f(x_{\text{query}}, D_{\text{train}})$ that has access to $D_{\text{train}}$ at inference time. The dataset is composed of $n$ datapoints $x^{(i)} \in \mathbb{R}^a$, each with $a$ attributes and $l$ labels $y^{(i)} \in \mathbb{R}^l$. Our proposed models operate over an embedding $\mathbf{X} \in \mathbb{R}^{n \times (a+l) \times d}$ of the training set, obtained by projecting each label and attribute into a dense $d$-dimensional embedding.

**High-Level Overview**   We introduce the Non-Parametric State Space Model (NPSSM), an architecture that produces a distributed representation of the training data that is useful for downstream tasks. Our method starts with an initial embedding $\mathbf{X}$ of a dataset and iteratively updates this representation by applying a sequence of alternating SSM layers across the datapoints and across the attributes. These SSM layers are analogous to attention mechanisms over datapoints and attributes used by the Axial Transformer and subsequent work (Ho et al., 2019; Kossen et al., 2021; Rao et al., 2021), but

are more computationally efficient and support longer contexts. These alternating layers allow the model to capture relationships between the attributes for each datapoint, and relationships between the datapoints along a given attribute. For notational simplicity, we will assume that $x_{\text{query}}$ has been concatenated to $\mathbf{X}$ and we will omit the batch dimension.

The model is trained by masking a percentage of $\mathbf{X}$ and then reconstructing the masked elements by optimizing the negative log-liklihood. This effectively defines a probabilistic model $p(x\|D_{\text{train}})$.

## 3.1 ARCHITECTURE

Each NPSSM layer takes as input an embedding $\mathbf{X}$ of $D_{\text{train}}$ and returns $\hat{\mathbf{X}} \in \mathbb{R}^{n \times (a+l) \times d}$. A layer has two components: an SSM along attributes (SSM$_{\text{attr}}$) and a SSM along datapoints (SSM$_{\text{data}}$).

$$\hat{\mathbf{X}} = \text{SSM}_{\text{data}}(\text{SSM}_{\text{attr}}(\mathbf{X}))$$

**Attribute-Level State-Space Layers**   To model the interactions between individual attributes and labels, we apply a state-space model along the attributes and labels of each individual datapoint within each dataset, $\text{SSM}_{\text{attr}} : \mathbb{R}^{n \times (a+l) \times d} \mapsto \mathbb{R}^{n \times (l+a) \times d}$. The SSM$_{\text{attr}}$ layer can be instantiated with any arbitrary SSM; in this work we used a bidirectional version of Mamba (BiMamba) from Schiff et al. (2024). To achieve bidirectionality we apply a Mamba operator along the length dimension of $\mathbf{X}$, and we then apply a Mamba operator on $\mathbf{X}$ reversed along the length dimension. Let reverse$(M, \text{dim})$ reverse matrix $M$ along dimension dim; we then write:

$$\begin{aligned}
\text{SSM}_{\text{attr}}(\mathbf{X}) &= \text{BiMamba}(\mathbf{X}) \\
&= \text{Mamba}(\mathbf{X}) + \text{reverse}(\text{Mamba}(\text{reverse}(\mathbf{X}, 1), 1)
\end{aligned}$$

As in Schiff et al. (2024), the "forward" and "reverse" Mamba operators share the parameters of the input and output projection matrices to reduce the total number of trainable parameters.

**Datapoint-Level State-Space Layers**   To model the interactions between individual datapoints, we apply a SSM along the attribute dimension to model the interactions between each datapoint at every attribute. Note that this process is equivalent to applying an SSM$_{\text{attr}}$ layer, but permuting the datapoint ($n$) and attribute ($a + l$) dimensions of $\mathbf{X}$ to form matrix $\mathbf{X}' \in \mathbb{R}^{(a+l) \times n \times d}$.

$$\begin{aligned}
\text{SSM}_{\text{data}}(\mathbf{X}') &= \text{BiMamba}(\mathbf{X}') \\
&= \text{Mamba}(\mathbf{X}') + \text{reverse}(\text{Mamba}(\text{reverse}(\mathbf{X}', 1), 1)
\end{aligned}$$

**Non-Parametric State Space Models**   A single layer of a NPSSM is composed of a Attribute Level State Space Layer (SSM$_{\text{attr}}$) and a Datapoint Level State Space (SSM$_{\text{data}}$) layer applied consecutively as described in Algorithm 1. A full NPSSM model is composed of multiple of these layers chained together, trained on a MLM objective $\mathcal{L}^{\text{MLM}}$ to reconstruct the input $\mathbf{X}$ with an optional auxilliary supervised loss ($\mathcal{L}^{\text{aux}}$) computed on a subset of labeled datapoints in the context set.

---

**Algorithm 1** NPSSM Block Layer

---

**Input:** $\mathbf{X} : (N, (A + L), D)$
  $\hat{\mathbf{X}} \leftarrow \text{SSM}_{\text{attr}}(\mathbf{X})$
  $\mathbf{X}' \leftarrow \hat{\mathbf{X}}.\text{permute}(1,0,2))$                              $\triangleright$ ((A+L), N, D)
  $\hat{\mathbf{X}} \leftarrow \text{SSM}_{\text{data}}(\mathbf{X}')$
  $\hat{\mathbf{X}} \leftarrow \hat{\mathbf{X}}.\text{permute}(1,0,2))$                              $\triangleright$ (N, (A+L), D)
  **return** $\hat{\mathbf{X}}$

---

**Supervised Learning Architecture**   For supervised learning problems, we have an input $x_{\text{query}}$ where the $l$ labels are masked out but the $a$ attributes remain unmasked, in addition to the masked embedded matrix $\mathbf{X}$. The input $x_{\text{query}}$ is concatenated to $\mathbf{X}$ and an additional loss $\mathcal{L}^{\text{aux}}$ is calculated on the $l$ labels, for example a BCE loss, $L_2$ loss, etc. The full loss is therefore $\mathcal{L}^{\text{total}} = (1 - \lambda)\mathcal{L}^{\text{MLM}} + \lambda\mathcal{L}^{\text{aux}}$ for some weight $\lambda$ (see below).

**Multiple Sequence Alignment State Space Models**    NPSSMs lend themselves naturally to Multiple Sequence Alignments (MSA)s. Each sequence is an individual datapoint and each attribute/token is already aligned. Each sequence may have any labels or additional attributes appended at the end of each sequence allowing for a straightforward processing of the MSA input as in Notin et al. (2023b).

### 3.2 TRAINING

**Supervised Learning**    The NPSSM model is trained optimizing the weighted sum of an MLM and supervised objective: $\mathcal{L}^{\text{total}} = (1 - \lambda)\mathcal{L}^{\text{MLM}} + \lambda\mathcal{L}^{\text{aux}}$ for some weight $\lambda$. To compute the reconstruction loss $\mathcal{L}^{\text{MLM}}$, we mask a portion of all the elements of $\mathbf{X}$ ($10\%$ for imputation). For example, for discrete data, we compute the cross-entropy loss of the predicted tokens with the true token in $\mathbf{X}$. More generally, we can append to the NPSSM a final layer that predicts the parameters of a probability distribution over $\mathbf{X}$ or over $x_{\text{query}}$; this defines a probabilistic model $p(x_{\text{query}}|\mathbf{X})$ that is naturally amenable to maximum-likelihood training.

The auxiliary loss $\mathcal{L}^{\text{aux}}$ is computed based on the masked labels of $x_{\text{query}}$ and a subset of datapoints in $\mathbf{X}$ that have the labels masked. For our imputation experiments, the auxiliary loss is a weighted version of the cross entropy loss.

**Meta-Learning**    For meta-learning, we want to learn some function $y_{\text{query}} = f(x_{\text{query}}; D_{\text{train}})$ that generates $y_{\text{query}}$ given an input $x_{\text{query}}$ and a context dataset $D_{\text{train}}$. Models like NPSSM lend themselves to this application. To train a model for this setting, we first construct a meta-learning set $\{D_{\text{train}}^{(i)}\}_{i=1}^{H}$. In our motivating example of genotype imputation, each individual dataset corresponds to a different region of the genome, and the model is trained to be able to impute arbitrary regions of the genome, even for samples regions that were not seen during training.

## 4 EXPERIMENTS

### 4.1 PROTEIN ANALYSIS

Protein NPT (PNPT) is a non-parametric model specifically designed to be used on Deep Mutational Scanning (DMS) assays, protein assays that experimentally measure a varied set of functional properties for a large number of protein sequences (Notin et al., 2023b;a). These assays are often highly structured, taking the form of a MSA with functional labels appended to each sequence. This organized structure makes them a prime candidate for the application of non-parametric models: given some protein of interest and a DMS assay of related protein, the model should belable to use the information from the DMS assay to predict some properties of the protein of interest.

Table 1: Results averaged over 13 DMS tasks. Each task was averaged over 5 seeds.

**Experimental Setup**    We follow a similar set up to PNPT: we focus on the single mutant property prediction task where we predict the effect of a single mutation on the fitness of a given protein given a DMS assay containing the fitness for other single mutations of the same protein of interest (Notin et al., 2023b). We use the same 5-fold cross-validation scheme that Notin et al. (2023b) used, using their random cross-validation splits. We use the same hyperparmeter con-

| Method | $k$ | Spearman | Mem usage (Gb) |
|--------|------|----------|----------------|
|        | 1000 | 0.65     | 14.56          |
| PNPT   | 1500 | 0.64     | 18.66          |
|        | 2000 | OOM      | OOM            |
|        | 1000 | 0.65     | 7.77           |
| NPSSM  | 1500 | 0.65     | 9.54           |
|        | 2000 | 0.65     | 12.07          |

figuration as PNPT when possible, with the notable exception that we experiment with larger context set sizes during training and evaluation. Additionally, due to computational constraints we limit the PNPT and NPSSM NPSSM models to 1 Million parameters each, down from $\sim 3.5$ Million in Notin et al. (2023b) and we consider 13 tasks where the PNPTs could fit in memory on NVIDIA 3090 24GB GPUs.

### 4.1.1 RESULTS

We show that Non-Parametric State Space Models achieve competitive performance with existing attention based models in terms of Spearman's rank correlation while utilizing between 1.5 and 2 times less GPU memory (Table 1). For this particular setup, increasing the context set size from PNPTs original work $k = 1000$ had little effect in terms of performance for either model but it resulted in a significant increase in GPU memory consumption for PNPT.

## 4.2 GENOTYPE IMPUTATION

Genotype imputation is an important component of many genomic analysis and a natural application of meta-learning. Given some binary observed variants $x_{\text{query}} = (x_1, \ldots, x_t)$ and unobserved variants $y_{\text{query}} = (y_1, \ldots, y_u)$, the goal is to predict all $u$ unobserved variants using the $t$ observed variants. Generally, the observed variants $x$ are obtained by genotyping an individual using a relatively cheap DNA microarray at relatively few positions, that is $t << u$. The current methods for genotype imputation all follow the same general setup, we assume that at inference time we have access to some reference set $H_{\text{ref}} \in \mathbb{1}^{k \times (u+t)}$ that contains $k$ fully sequenced individuals and some new individual $x_{\text{query}}$ that is only genotyped at some subset of variants of size $t$.

To solve this problem, we want some function $y_{\text{query}} = f(x_{\text{query}}; H_{\text{ref}})$. The current approaches are fully non-parametric, they assume that $x_{\text{query}}$ is a mosaic of the samples in $H_{\text{ref}}$, so the problem can be solved by finding the most probable path through the samples in $H_{\text{ref}}$ that result in $x_{\text{query}}$ using an HMM from Li & Stephens (2003). This setup lends itself naturally to meta-learning, in practice $H_{\text{ref}}$ contains an arbitrary number of haplotypes and $H_{\text{ref}}$ covers an arbitrary section of the genome. By training over a meta training set consisting of a variety of $H_{\text{ref}}$'s, we can train a model that should work for any arbitrary $(x_{\text{query}}, H_{\text{ref}})$ pair like existing methods do.

**Experimental Setup** We follow the setup of Rubinacci et al. (2020) and split up the 1000 Genomes dataset (The 1000 Genomes Project Consortium, 2015) into a training set with 4388 sequences (haplotypes), a validation set with 516 haplotypes, and a test set on 104 haplotypes. We consider chromosome 20, and use the IlluminaOmiExpress-24 microarry to define the typed and untyped variants. We subset chr20 to a region of 9159 contiguous untyped variants ($\sim 1.3\%$ of chr20) into 92 blocks with 100 untyped variants and the closest 200 typed variants per block. Each datapoint $(x_{\text{query}}, y_{\text{query}})$ is a single block for one of the samples in the 4388 training sequences, leading to $92 \times 4388 = 403696$ $(x_{\text{query}}, y_{\text{query}})$ data points. Each $x_{\text{query}}$ has an associated context set $H_{\text{ref}}$ composed of $k \le 4387$ datapoints $(x_{\text{query}}, y_{\text{query}})$ from the training set on that same block. We subset each $H_{\text{ref}}$ to the $k$ closest haplotypes by $L_1$ distance on $x_{\text{query}}$ for each block. This results in a meta-dataset of 403696 datasets where each individual dataset is composed of $(x_{\text{query}}^{(0)}, y_{\text{query}}^{(0)}, H_{\text{ref}} = \{(x_{\text{query}}^{(i)}, y_{\text{query}}^{(i)})\}_{i=1}^k)$ for the $k$ datapoints on the same block with the smallest $L_1$ distance to $x_{\text{query}}^{(0)}$. The eval and test datasets also construct $H_{\text{ref}}$ from the 4388 training haplotypes to ensure that no haplotypes from the validation or test set ever appear in $H_{\text{ref}}$. We compare to two industry standard tools based on the Li & Stephens (2003) HMM, Beagle (Browning et al. (2018)) and Impute (Rubinacci et al. (2020)), to a logistic regression basline, and to two non-parametric methods, KNN and MSA-Transformer Rao et al. (2021); Kossen et al. (2021). Full hyperparameter configurations for NPSSM and MSA Transformer are in Section A.1.

**Results** We report the $r^2$ over all 9159 variants on 516 haplotypes in Table 2 and list the hyper parameters in Section A.1. NPSSM is able to achieve SOTA performance, outperforming existing HMM based methods. We additionally report the $r^2$ binned by the Minor Allele Frequency of each untyped variant in Figure 1 where we show that NPSSM is able to match or exceed the performance of existing methods, even for rare SNPs.

## 4.3 ABLATIONS

### 4.3.1 GENOTYPE IMPUTATION GENERALIZABILITY

To asses how well the models from 4.2 generalize to unseen data, we take those models trained on 1.3% of chromosome 20 and we evaluate them on the entirety of chromosome 20 (670k variants). Even though this data is on the same haplotypes and on the same chromosome, the majority of the

Table 2: Imputation performance ($r^2$) evaluated on 9159 untyped variants (dev set) from 516 haplotypes on chromosome 20.

| Class | Method | $k$ | $r^2$ | $(1 - r^2)$ |
|---|---|---|---|---|
| Traditional ML | KNN | 4388 | $0.758 \pm 0.005$ | $0.242 \pm 0.005$ |
| | LR | NA | $0.882 \pm 0.006$ | $0.118 \pm 0.006$ |
| HMM | Beagle (Browning et al. (2018)) | 4388 | $0.943 \pm 0.002$ | $0.057 \pm 0.002$ |
| | Minimac4 (Das et al. (2016)) | 4388 | $0.931 \pm 0.003$ | $0.069 \pm 0.003$ |
| | Impute (Rubinacci et al. (2020)) | 4388 | $0.935 \pm 0.003$ | $0.065 \pm 0.003$ |
| Non-Parametric Models | SPIN | 4388 | $0.868 \pm 0.010$ | $0.132 \pm 0.010$ |
| | MSA Transformer | 650 | $0.946 \pm 0.002$ | $0.054 \pm 0.002$ |
| | NPSSM | 2000 | $0.950 \pm 0.002$ | $0.050 \pm 0.002$ |

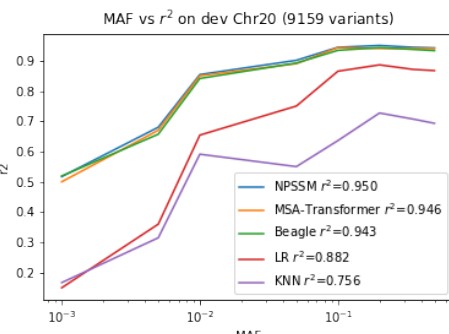

Figure 1: $r^2$ on 9159 untyped variants in chromosome 20 binned by MAF frequency.

data is complete unseen, testing the capability of the model to adapt to what is essentially data points outside the distribution of what the model was trained on.

**Results**    Table 3 shows that NPSSM is still able to outperform existing methods on the entirety of chromosome 20, even when the model was only trained on $1.3\%$ of chromosome 20.

### 4.3.2    SCALING WITH CONTEXT SIZE

In Figure 2, we show the scaling of multiple methods with respect to the size of the context set $k$. As $k$ increases, a naive method like KNN sees a brief increase followed by a decrease in performance. More sophisticated non-parametric methods monotonically increase in $r^2$ as $k$ increases. However, existing methods based on quadratic self attention run out of memory for $k \geq 650$ on a 48 GB card, limiting the maximum context set size they handle and consequently their performance. By contrast, NPSSM scales linearly with the size of the context set, allowing for a larger context set and consequently increased performance. Figure 5 showcases the quadratic scaling of traditional transformer architectures and the linear scaling NPSSM achieves. This better scaling allows for performance gains by allowing a larger context set on the same amount of GPU memory compared to previous work.

### 4.3.3    OBJECTIVE

There are four main knobs to tune the objective $\mathcal{L}^{\text{total}}$, the masking strategy of $\mathcal{L}^{\text{MLM}}$, the masking rate of $\mathcal{L}^{\text{MLM}}$, the auxlliary loss chosen $\mathcal{L}^{\text{aux}}$, and the weight factor $\lambda$ weighing the two losses.

**Masking Strategy**    Existing works such as Notin et al. (2023b) masks individual tokens at random with proba-

Table 4: Performance using different masking strategies for $\mathcal{L}^{\text{MLM}}$.

| Masking Strategy | $r^2$ |
|---|---|
| Tokens | 0.635 |
| Datapoints | 0.934 |
| Attributes | 0.613 |

Table 3: Imputation performance ($r^2$) evaluated on 670370 untyped variants (full set) from 516 haplotypes on chromosome 20. Trained models were only trained on the dev set (9159 untyped variants).

| Class | Method | $r^2$ |
|---|---|---|
| HMM | Beagle (Browning et al. (2018)) | $0.955 \pm 0.004$ |
| | Impute (Rubinacci et al. (2020)) | $0.952 \pm 0.004$ |
| Non-Parametric Models | MSA Transformer | $0.955 \pm 0.003$ |
| | NPSSM | $0.958 \pm 0.003$ |

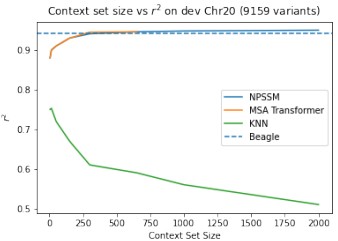 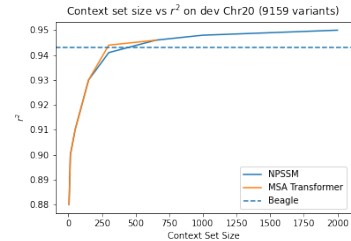

Figure 2: $r^2$ by context set size on 9159 untyped variants from chr20.

bility $p_{\text{mask}}$ for reconstruction, but previous work Rastogi et al. (2023) observed that different masking strategies had a pronounced impact on the performance of the models. Table 4 shows the effect of various masking strategies by masking out random tokens, masking out random spans of each datapoint (row), and masking out random spans of each attribute (column).

**Masking Rate** The masking rate $p_{\text{mask}}$ is an important parameter for training MLM models, particularly if the masking strategy is more involved than masking out individual tokens. In Figure 3 we report the $r^2$ of a model masking out spans of datapoints.

Table 5: Performance after training on the MLM objecive $\mathcal{L}^{\text{MLM}}$ only, the supervised objective $\mathcal{L}^{\text{aux}}$ only, or $\mathcal{L}^{\text{total}}$ with a fixed weight $\lambda$ or an annealed weight.

| Training Objective | $r^2$ |
|---|---|
| $\mathcal{L}^{\text{MLM}}$ | 0.791 |
| $\mathcal{L}^{\text{aux}}$ | 0.934 |
| $\mathcal{L}^{\text{total}}, \lambda = 0.9$ | 0.939 |
| Annealed $\mathcal{L}^{\text{total}}$ | 0.940 |

**Auxilliary Loss** Imputation methods are often judged by their performance grouped by Minor Allele Frequency (MAF), a metric that describes the proportions of labels for each variant/attribute. In Figure 4, we explore different variants of $\mathcal{L}^{\text{aux}}$ included a standard cross entropy loss (CEL), a MAF-weighted CEL, and CEL annealed between both. We find that the MAF weighted loss is better at predicting rare variants, but worse on the common variants than the standard CEL. Annealing between both the losses results in the best performance over all MAF buckets.

**Combined Objective** Table 5 shows the performance of a model trained purely on the reconstruction objective $\mathcal{L}^{\text{MLM}}$, trained only on the supervised objective $\mathcal{L}^{\text{aux}}$, or on combined objective $\mathcal{L}^{\text{total}} = (1 - \lambda)\mathcal{L}^{\text{MLM}} + \lambda\mathcal{L}^{\text{aux}}$ for a fixed $\lambda$ or an annealed $\lambda$ starting with $\lambda = 1$ and following a cosine annealing schedule to $\lambda = 0.01$. Training on a combined loss $\mathcal{L}^{\text{total}} = (1 - \lambda)\mathcal{L}^{\text{MLM}} + \lambda\mathcal{L}^{\text{aux}}$ yields better performance than optimizing on either objective individually.

### 4.3.4 CONTEXT LENGTH EXTENSION

As shown in Section 4.3.2, the number of datapoints in the context set $k$ has a large influence on the performance for genotype imputation. In Table 6 we explore the *post hoc* expansion of the context set size during evaluation, that is we train on some context set size $k$ and then during evaluation we increase the context set size without any additional training.

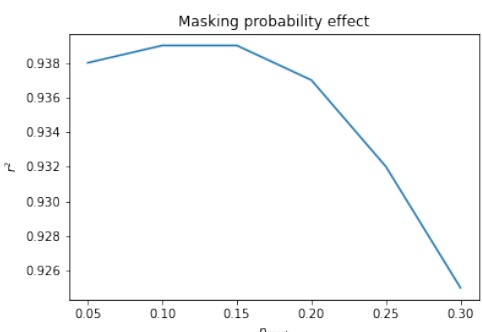

Figure 3: $r^2$ for different settings of $p_{\text{mask}}$.

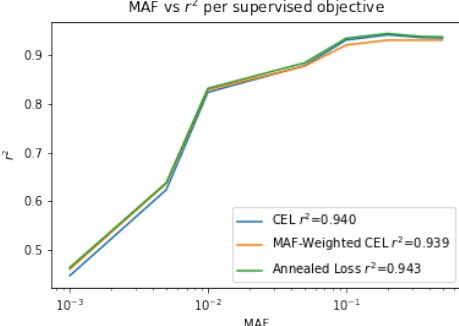

Figure 4: $r^2$ on 9159 untyped variants in chromosome 20 binned by MAF frequency for different $\mathcal{L}^{\text{aux}}$ choices, a standard cross entropy loss (CEL), a Minor Allele Frequency (MAF)-weighted CEL, and a loss annealed between the both the weighted and unweighted losses.

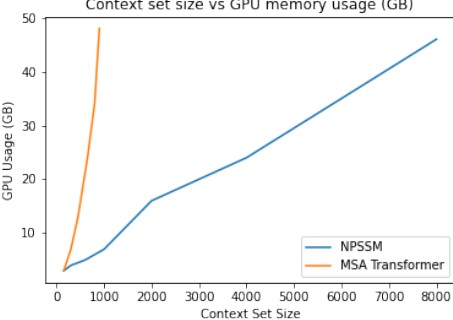

Figure 5: GPU memory usage for different context set sizes $k$ during evaluation where each context point is of length 600 (i.e. $H_{\text{ref}} \in \mathbb{R}^{k \times 600}$) for various $k$'s.

For NPSSM models, increasing $k$ at evaluation time provides a modest increase in performance, but not as large an increase as training on the expanded context size instead. The MSA Transformer model however showed the opposite effect, where increasing the context set size during evaluation resulted in a decrease in performance, counter to the intuition that at worst expanding the context set size would result in no change in performance. Note that MSA Transformer treats the datapoints as a set, there is no positional embedding used for the datapoints as in the original work by Rao et al. (2021).

Table 6: Imputation performance ($r^2$) evaluated on 9159 untyped variants from 516 haplotype on chromosome 20, extending the number of haplotypes seen during evaluation.

| Method | $k$ during training | $k$ during eval | eval $r^2$ |
|---|---|---|---|
| MSA Transformer | 150 | 150 | 0.9407 |
| | | 300 | 0.9367 |
| | | 650 | 0.9156 |
| | 300 | 300 | 0.9440 |
| | | 650 | 0.9429 |
| | 650 | 650 | 0.9461 |
| NPSSM | 150 | 150 | 0.9428 |
| | | 300 | 0.9444 |
| | | 650 | 0.9445 |
| | 300 | 300 | 0.9474 |
| | | 650 | 0.9489 |
| | | 1000 | 0.9489 |
| | 650 | 650 | 0.9495 |
| | | 1000 | 0.9499 |
| | | 2000 | 0.9501 |

## 5 DISCUSSION AND RELATED WORK

**Comparisons to NPTs** NPSSMs are generally a drop in replacement for models like MSA-Transformer or Non Parametric Transformer (Rao et al., 2021; Kossen et al., 2021; Notin et al., 2023b). Each of these methods make similar assumptions on the structure of the input data, therefore they can be applied to similar problems. However there are still some capabilities of attention based models that do not have an analogue in SSMs, for example Rao et al. (2021) used attention maps for contact predictions, and they had success with forcing each datapoint to share the same attention matrix (tied row attention). While NPSSMs have shown competitive results with much better scaling on the genotype imputation task, there are certain scenarios like these where SSM based non-parametric models are not equivalent out of the box.

**Comparisons to HMMs** Models like Beagle and Impute (Browning et al., 2018; Rubinacci et al., 2020) that are based off of the Li & Stephens (2003) HMM have been the standard for genotype imputation. However, Non-parametric models like NPSSM are now capable of matching or exceeding their performance, even on a whole chromosome (Table 3) due to their favorable scaling properties. In addition, existing HMM imputation methods are restrictive. Adding in additional auxiliary for imputation like Notin et al. (2023b) did for proteins is difficult for HMMs, but when using a non-parametric model it could be as simple as adding some additional columns to $\mathbf{X}$ and adjusting $\mathcal{L}^{\text{aux}}$.

**Conclusion** We present NPSSMs, a drop in replacement for existing attention based non-parametric architectures that scale *linearly* in the size of the context set. We show that on genotype imputation, we are able to outperform existing models (Table 2), and that a large contributing factor to performance is the size of the context set we are able to accommodate (Figure 2).

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

## A    APPENDIX

### A.1    HYPERPARAMETERS

Table 7 contains the hyperparameters used for NPSSM (700k params) and Table 8 contains the hyperparameters used for MSA Transformer (700k) params.

Table 7: Hyperparameters for 700k paramater NPSSM

| | |
|---|---|
| Embedding dim $d$ | 156 |
| Number of NPSSM layers | 2 |
| Initial learning rate | $1e - 3$ |
| $p_{mask}$ | 0.10 |
| optimizer | Adam |
| effective batch size | 128 |
| Starting $\lambda$ | 1.0 |
| $\lambda$ annealing schedule | linear |
| Ending $\lambda$ value | 0.5 |

Table 8: Hyperparameters for 700k paramater MSA Transformer

| | |
|---|---|
| Embedding dim $d$ | 100 |
| Num attention heads | 10 |
| Number of MSA-Transformer layers | 4 |
| Activation Dropout | 0.1 |
| Attention Dropout | 0.1 |
| Initial learning rate | $6e - 4$ |
| $p_{mask}$ | 0.10 |
| optimizer | Adam |
| effective batch size | 128 |
| Starting $\lambda$ | 1.0 |
| $\lambda$ annealing schedule | linear |
| Ending $\lambda$ value | 0.5 |
| Datapoint Positional Embedding | None |
| Tied Row Attention | True |

### A.2    LAYER ABLATIONS

To investigate the effects of each individual layer, we construct five identical models with the same hyperparameters as in 7 with a context size of $k = 150$. We then train these models for 100,000 steps and and evaluate the results on the same set of data as in Table 2. We include these results in Table 9. We observe that permuting the order of the layers results in a noticeable decrease in performance, indicating that the ordering of the layer application is an important consideration for these class of models. We also observe that flattening the input, that is concatenating all 150 back to back to form a single sequence as an input, performance significantly worse than either of the models that operates on the MSA, achieving only $88\%$ of the performance that the base model achieves. The final two ablations correspond to applying the layers only over the attributes/rows (i.e. a standard SSM) and only over the data points/columns. The $SSM_{attr}$ model performance similarly to the flattened version even though the $SSM_{attr}$ model does not have access to any external context set. This might be indicative that the performance of the Flattened version of the model was impacted due to the increased difficulty of using the context set without leveraging the MSA structure. The $SSM_{data}$

model observes only, which for this task without any relevant context likely limits the model to performing some sort of weighted average. This model converged fairly quickly (under $10,000$ steps) and performs similarly to the KNN baseline from Table 2 (0.758).

## A.3  ADDITIONAL DATASETS

We additionally report performance of Beagle and the two best Non-Parametric Models on a different chromosome of 1k genomes, chr14 (Table 10) and on a different dataset HapMap (The International HapMap 3 Consortium (2010)) (Table 11).

Table 9: Imputation Performance ($r^2$) evaluated on 9159 variants from 516 haplotypes on chromosome 20. Models were trained for 100,000 steps, each with the same model configuration. Flattened is taking the input $H_{\mathrm{ref}}$ and flattening it down into a single sequence (i.e. laying all the sequences in $H_{\mathrm{ref}}$ back to back.

| Model | $r^2 \pm \sigma$ |
|---|---|
| (Base) $\mathrm{SSM_{attr}} \mapsto \mathrm{SSM_{data}}$ | $0.9406 \pm 0.0026$ |
| $\mathrm{SSM_{data}} \mapsto \mathrm{SSM_{attr}}$ | $0.9308 \pm 0.0029$ |
| Flattened | $0.8241 \pm 0.0067$ |
| $\mathrm{SSM_{attr}}$ Only | $0.8206 \pm 0.0074$ |
| $\mathrm{SSM_{data}}$ Only | $0.7783 \pm 0.0049$ |

Table 10: Imputation performance ($r^2$) evaluated on $\sim 19,369$ untyped variants (dev set) from 516 haplotypes on chromosome 14. The Non-Parametric Models were trained on chromosome 20.

| Class | Method | $k$ | $r^2$ |
|---|---|---|---|
| HMM | Beagle (Browning et al. (2018)) | 4388 | $0.964 \pm 0.002$ |
| Non-Parametric Models | MSA Transformer | 650 | $0.963 \pm 0.002$ |
| | NPSSM | 2000 | $0.967 \pm 0.002$ |

Table 11: Imputation performance ($r^2$) evaluated on 962 untyped variants from 400 haplotypes from Hapmap on chromosome 14. The Non-Parametric Models were trained on chromosome 20 on 1000 Genomes.

| Class | Method | $k$ | $r^2$ |
|---|---|---|---|
| HMM | Beagle (Browning et al. (2018)) | 1828 | $0.891 \pm 0.008$ |
| Non-Parametric Models | MSA Transformer | 650 | $0.921 \pm 0.007$ |
| | NPSSM | 1828 | $0.919 \pm 0.007$ |

