# OpenReview forum: "Non-Parametric State-Space Models Over Datapoints and Sequence Alignments"
_ICLR.cc/2025/Conference — Submitted to ICLR 2025_

### Official Review · Reviewer_jFin · 2024-10-27

**Soundness:** 4
**Presentation:** 3
**Contribution:** 3
**Rating:** 6
**Confidence:** 4

**Summary:**

In this work, the authors apply SSMs, an efficient sequence modeling technique, to non-parametric transformers. The experiments validate that the Non-Parametric State Space Models (NPSSM) attain similar performance to exising attention-based methods.

**Strengths:**

- The method is well-motivated. Given that current non-parametric models typically rely on the self-attention mechanism, replacing attention with SSMs is a reasonable approach.
- This submission validates the effectiveness of SSMs in modeling, contributing to a better understanding of SSMs in the field.

**Weaknesses:**

- (**major**) The paper includes an acknowledgment section, at the end of the submission, which potentially reveals author information and violates the double-blind review policy.
- Since SSM can be viewed as an efficient sequential modeling technique, the authors should compare their method with other efficient attention algorithms, including linear attention works.
- It should help if authors provide a formal formulation of \mathcal{L}_{MLM} and \mathcal{L}_{Aux}. I'm still a little confused about the method pipeline.

**Questions:**

Can this approach be applied to other tasks, such as language tasks?

**Details Of Ethics Concerns:**

The paper includes an acknowledgment section, which potentially reveals author information and violates the double-blind review policy.

---

> ### Author Response · Authors · 2024-11-22
> **Response to Reviewer jFIN**
>
> Thank you for your feedback!
> ## Concern 1: Acknowledgements
> To our knowledge, we did not include an Acknowledgements section in the posted version of the paper.
> ## Concern 2: Additional Baselines
> * We **add in SPIN as a baseline, a non-parametric architecture using linear attention**. SPIN was also evaluated on a variant of genotype imputation and was compared against other attention mechanisms and showed competitive or the best performance on their tasks. We incorporate SPIN into our workflow and evaluate SPIN on over 9k untyped variants in Table 2. Note that their tasks only considered a relatively small number of SNPs at a time (O(100)) where we consider a much larger problem more representative of real world use cases. This may account for performance discrepancies between their reported values and this work.
> * We additionally add in another HMM baseline, Minimac and update results reporting. Full details are in the General comment, but briefly:
>   * We bootstrap our results to get confidence bounds
>   * We report the error $(1-r²)$ to make the performance gains even more apparent (e.g a $\\sim 12\\%$ decrease in error rate for NPSSM over the best existing HMM).
>   * We add in more datasets (Table 10 on a different Chromosome and Table 11 on a different dataset and different chromosome).
> ## Table 2
> | Class     | Method             | $k$                   | $r^2$                | $(1-r²)$
> |---------------|------------|-------|----------------------|-----|
> | Traditional ML        | KNN      | 4388       | $0.758 \\pm 0.005$    |  $0.242 \pm 0.005$ |
> |      | LR     | NA     | $0.882 \\pm 0.006$    | $0.118 \pm 0.006$ |
> | HMM    | Beagle      | 4388           | $0.943 \\pm 0.002$    | $0.057 \pm 0.002$ |
> |      | Minimac4 | 4388           | $0.931 \\pm 0.003$    | $0.069 \pm 0.003$ |
> |      | Impute     | 4388            | $0.935 \\pm 0.003$ | $0.065 \pm 0.003$ |
> | Non-Parametric Models | SPIN      | 4388    | $0.868 \\pm 0.010$    | $0.132 \pm 0.010$ |
> |       | MSA Transformer                  | 650     | $0.946 \\pm 0.002$  | $0.054 \pm 0.002$ |
> |      | NPSSM                      | 2000    | $0.950 \\pm 0.002$  | $0.050 \pm 0.002$ |
>
>
> ## Concern 3: Unclear Loss Formulations
> Let's consider as an example the imputation problem ignoring batch size to clean up the notation. Our input is $X \\in \\mathbb{R}^{N \\times (A+L)}$ where the first row ($X[0,\\ldots]$ = $[x_1, x_2, \\text{<MASK>}, \\ldots, x_{a+l}]$) is our partially masked query sequence and the rest is our  partially masked context set. Our output is $\\hat{Y} \\in
>  \\mathbb{R}^{N \\times (A+L) \\times V}$ for some vocabulary of length $V$ and our labels are $Y \\in \\mathbb{R}^{N \\times (A+L)}$.
>  Our MLM loss is just a standard cross entropy loss calculated only on the reference set $\\mathcal{L}^{\\text{MLM}} = \\text{CEL}(\\hat{Y}[1:,\\ldots], Y[1:, \\ldots])$, and the Aux loss is a weighted version of the cross entropy loss calculated only on the query sequence (dropping the weight term for simplicity),
> $\\mathcal{L}^{\\text{Aux}}=\\text{CEL}_{\\text{weighted}}(\\hat{Y}[0,\\dots], Y[0, \\ldots])$ .
>
> Therefore the full loss would look like:
>  \\begin{align}
>  \\mathcal{L}^{\\text{total}} &= (1-\\lambda) \\mathcal{L}^{\\text{MLM}} + \\lambda\\mathcal{L}^{\\text{Aux}} \\\\
>  &= (1-\\lambda) \\text{CEL}(\\hat{Y}[1:,\\ldots], Y[1:, \\ldots]) +
>  \lambda \\text{CEL}_{\\text{weighted}}(\\hat{Y}[0,\\ldots], Y[0, \\ldots])\\\\
>  \\end{align}
>  for some weight parameter $\\lambda$. Conceptually, we index out the context set and calculcate a MLM loss over that, and we index out some sample/attribute of interest and calculcate a different loss over that (in this case a weighted cross entropy loss). In practice $\\mathcal{L}^{\\text{Aux}}$ can be any arbitrary loss, BCE, MSE, $L^1$, etc, as long as you index the appropriate portions of $\\hat{Y}, Y$ to calculate the loss over and reshape the outputs and targets as necessary (e.g. flattening $Y$ and $\\hat{Y}$ ). We hope this example cleared things up, but if not please let us know where the confusion is and we can attempt to address it.
> ## Questions
> Can this approach be applied to other tasks, such as language tasks?
> >Approaches like this could be applied to language tasks, but they require having a structured/aligned input. In biological sequences, alignments like MSAs have been commonly used for a long time and often occur during analysis, making them a natural use case for models of this sort. Natural language tasks that have a structure that lends itself to an alignment would be a viable task to apply this sort of model to, but if the task doesn't naturally have some sort of structure/alignment, this type of model may not be suitable for that task, leading to this architecture being of particular use in biological tasks.
>
> We hope that if this response adequately addressed the reviewers concerns, they will consider updating their score.

---

### Official Review · Reviewer_nAmg · 2024-10-31

**Soundness:** 3
**Presentation:** 3
**Contribution:** 2
**Rating:** 5
**Confidence:** 4

**Summary:**

This work proposes an SSM approach to non-parameteric learning. i.e., the setting where the dataset itself is used as input to the model, rather than learning a model based on the training data and then discarding the data. This setting is particularly advantageous for meta-learning where one seeks a model for new datasets.
In previous work, transformers have been proposed for this task (e.g., MSA transformer and NP transformers, mentioned in the paper). The current paper proposes to instead use state space models for this, motivated by their ability to handle longer contexts with less memory requirements.
On the one hand, the idea is reasonable, and the empirical results confirm that performance is competitive and does allow use of longer contexts. On the other hand, it seems like the novel component here is mostly replacing transformers with SSM, but the other conceptual aspects of using it for meta-learning (e.g., various masking losses) have been introduced in the previous NP transformer works.

**Strengths:**

As mentioned above, it is good to see that the advantages of SSM are also manifested in this setting.

**Weaknesses:**

It seems like conceptually the approach largely follows that of NP transformers, and thus the main contribution is to show that state space models are a viable alternative in this case. If the main claim is empirical, I would expect more extensive experiments and clearer gains.

**Questions:**

1. For Table 3, was the HMM trained on chromosome 20?
2. Typo: “Each these methods “ -> “Each of these methods “
3. HMMs seem to have similar performance to the non-parameteric methods in Table 2 (differences seem small and not statistically significant unless you show otherwise).. Can you comment on why HMM is an inferior solution in this case?
4. Another natural baseline, which I don't see but maybe I missed, is to just train on the target dataset D_C, with an SSM model (but still perform some pre-training with masking).

---

> ### Author Response · Authors · 2024-11-22
> **Response to Reviewer nAmg - Part 1**
>
> We thank the reviewer for their feedback
> ### Concern 1:  Novelty
>
> Our novelty lies in the following contributions:
> * We show for the first time that state-space models can be applied to a new task: non-parametric modeling of datapoints and sequence alignments. This expands the scope of tasks that can be addressed with SSMs.
> * We show that SSMs over MSAs are more scalable and performant than transformers over MSAs. The MSATransformer has been immensely important in biological applications (e.g., protein LMs, AlphaFold); improving it has potential for high impact. In that regards, our novelty relative to Mamba is analogous to the novelty of MSATransformer relative to the transformer.
> * We additionally derive a novel probabilistic formulation of these models and a meta-learning objective that goes beyond classical masked training of MSA models. We report that this training is crucial to solve the imputation task (see Table 5, a model trained only on a MLM objective version achieves only $85\\%$ of the performance our full setup achieves ).
> * While our architecture builds on top of BiMamba, producing a performant model requires solving a number of technical challenges. We report in the following sections the effect of these changes.
> * Designing a loss function (Figure 4)
> * Annealing schedules (Table 5)
> * Masking strategies (Table 4, Figure 3)
>
>  Genotype imputation has largely worked off the same model (Li and Stephens HMM) for over 2 decades now, and we show that these models have the potential to supplant these existing models. This however did not come from applying SSMs to the problem, a significant amount of engineering work was required.
>
>
> ### Concern 2: Limited Eval
>  We do the following to address our insufficient evaluation:
>
> 2a. **Add Baslines**:
>   1. We Add Semi-Parametric Inducing Point Networks **(SPIN) as a baseline**. This model is another non-parametric model, but it achieves linear scaling in the context size and was also evaluated in a Genotype Imputation setting against multiple other linear scaling architectures.
>   2. We Add in **Minimac4 as a baseline**, another HMM based method. This includes all commonly used methods for Imputation.
>
> 2b. **Add Datasets** We add further evaluation (Tables 10 and 11).
>   1.  (**Addition: New Dataset**) Table 11 (results on chr14 on HapMap genomes ) - We add results taking the models trained on a portion of chr20 on 1k genomes and evaluating them on chr14 of HapMap, an entirely different dataset. We split the HapMap dataset into a HapMap specific reference panel and a HapMap specific evaluation set. Given the different chromsome from training, and the context set being used, we argue there is no possiblity for any sort of data leakage and this adequately asses generalization. Note that this dataset is a worst case scenario for long context models (see Questions 3 for details), Yet we still observe HMM methods < NPSSM $\\approx$MSA-Transformer on the set of data.
>   2.  (**Addition: Different Chromosome**) Table 10 (results on chr14 on 1k genomes) - We add results taking the models trained on a portion of chr20 and evaluate them on chr14. We argue that chr14 shows entirely different patterns of relationships between variants and samples (e.g. even if 2 samples/haplotypes/datapoints were full copies of each on chr20, this need not hold for a different chromosome), and so chr14 should be considered as it's own dataset for the purposes of assesing generalizability. We observe HMMs < MSA Transformer < NPSSM on this set of data.
>
> 2c. **Add Statistical Significance**. We re-compute by bootstraping our results (10 bootstrap samples of $\sim 20\\%$ of the data by default). We then report the mean $r^2$ and $\\sigma$ of these trials and generally show HMM Methods < MSA-Transformer < NPSSM. For example, we now see that NPSSM is significantly better than HMMs in all experiments. In addition, we now also show the error rate$(1-r²)$ to better illustrate the performance gains. In Table 2 we see NPSSM is $\sim 12\\%$ better than the best HMM in terms of error rate (0.057 vs 0.050)

---

> ### Author Response · Authors · 2024-11-22
> **Response to Reviewer nAmg - Part 2 Questions**
>
> ## Questions
> 1. For Table 3, was the HMM trained on chromosome 20?
> > For Table 3- All HMM have parameters specific to the chromosome their being evaluated on. However, the different imputation methods are a mix of closed form parameters given some input and parameters determined by EM, details follow: The HMMs parameters for Impute and Beagle have a closed form based on the Li and Stephens model, and these parameters are determined for each chromosome. Impute and Beagle take in a “genetic map” that describes the structure of a given chromosome to calculate the specific parameters for their transition matrices at various locations along a given chromosome. Given this map, Impute and Beagle define all their transition matrices (not multiple transition matrices, these are example of non-homologous HMMs) .Minimac4 runs EM to fit the transition parameters based on the observed data, while Impute and Beagle don’t do any further fitting of the transition matrices. The emissions matrices are parametrized at the start for all models, but Beagle and minimac also run EM to further fit these parameters based on the observed chromosome data, although in general the parameter usually ends up in the 1e-3 to 1e-5 range.
> The Non-parametric models were trained on a subset of chr20, they do not have an equivalent of chromosome specific parameters, after training their weights remain fixed from chr20 for all results.
>
> Typo: “Each these methods “ -> “Each of these methods “
> >These corrections have been made, thank you!
>
> HMMs seem to have similar performance to the non-parameteric methods in Table 2 (differences seem small and not statistically significant unless you show otherwise).. Can you comment on why HMM is an inferior solution in this case?
> >As noted above, we now report confidence intervals for our results and show that some of the non-parametric models are significantly better than existing HMM methods.
> The exact algorithm differences from existing HMMs are indeed an interesting question, but are hard to assay. We hypothesize that the following are some of the possible reasons for the increased performance observed.
> > 1. Mutation Rate modeling (e.g. fixed mutation rate) - Existing HMM have an emissions matrix they use to model mismatches between a given observed variant in the query sequence and the reference sequences. The emissions matrix is parameterized by a single value (e.g. 1e-4 for Impute, ~1e-5 for Beagle), and while these methods are robust to reasonable err values [todo: cite impute 4, figure S4], the genome is known to have varying mutation rates along a given chromosome that existing HMMs likely do not model well. In Figure 1 we observe that non-parametric models outperform HMM based methods more at variants with higher MAFs, that is variants that are more likely to have a mutation are better modeled by non-parametric methods. Note that NPSSM still matches or exceeds HMM based methods at rare-variants too.
> > 2. Linear interpolation of untyped variants - Existing HMM methods only model copy states at the typed (observed) variants. The copy states for the unobserved variants are then interpolated between the bounding observed variants. Modeling these untyped variants as a convex combination of the bounding variants state probabilities ignores information that could be used for each untyped variant, such as the mutation rates for each of these untyped variants.
> > 3. In addition, one avenue that remains unexplored is that these non-parametric architectures admit richer inputs. For example, adding in recombination information, demographic information, or phenotype data as auxiliary inputs to this model could be as simple as adding in a few more columns as in PNPT. Incorporating  some of this information into existing HMM methods is non-trivial, and this may lead to even further performance gains or the ability to apply these models to a wide range of imputation problems. We look forward to studying these types of additions in follow-up work and further showing how this class of models supplants the existing HMMs for this use case.
>
> Another natural baseline, which I don't see but maybe I missed, is to just train on the target dataset D_C, with an SSM model (but still perform some pre-training with masking).
> >This baseline should correspond to the SSM_attr only ablation in Table 9, which is a SSM model with D_C used only as a training set and not added as an input during inference. Not having D_C at inference time seems to make the problem harder.
>
> We hope that if this response adequately addressed the reviewers concerns, they will consider updating their score.

---

> ### Author Response · Authors · 2024-11-22
> **Tables Referenced**
>
> ## Referenced Tables
> Below we attach some of the tables references in the rebuttal for convenience.
> ## Table 2
> | Class     | Method             | $k$                   | $r^2$                | $(1-r²)$
> |---------------|------------|-------|----------------------|-----|
> | Traditional ML        | KNN      | 4388       | $0.758 \\pm 0.005$    |  $0.242 \pm 0.005$ |
> |      | LR     | NA     | $0.882 \\pm 0.006$    | $0.118 \pm 0.006$ |
> | HMM    | Beagle      | 4388           | $0.943 \\pm 0.002$    | $0.057 \pm 0.002$ |
> |      | Minimac4 | 4388           | $0.931 \\pm 0.003$    | $0.069 \pm 0.003$ |
> |      | Impute     | 4388            | $0.935 \\pm 0.003$ | $0.065 \pm 0.003$ |
> | Non-Parametric Models | SPIN      | 4388    | $0.868 \\pm 0.010$    | $0.132 \pm 0.010$ |
> |       | MSA Transformer                  | 650     | $0.946 \\pm 0.002$  | $0.054 \pm 0.002$ |
> |      | NPSSM                      | 2000    | $0.950 \\pm 0.002$  | $0.050 \pm 0.002$ |
> ### Table 10 - Chr14 1k Genomes
>
> Imputation performance ($r^2$) evaluated on $19,369$ untyped variants (dev set) from $516$ haplotypes on chromosome 14. The Non-Parametric Models were trained on chromosome 20, and given the nature of the problem there should be no information flow between chromosomes.
> | Class         | Method      | $k$    | $r^2$     |
> |---------|----|------|-------|
> | HMM             | Beagle   | 4388       | $0.964 \pm 0.002$   |
> | Non-Parametric Models        | MSA Transformer      | 650     | $0.963 \pm 0.002$ |
> |         | NPSSM   | 2000      | $0.967 \pm 0.002$ |
>
> ### Table 11 - Chr14 HapMap
> Imputation performance ($r^2$) evaluated on $962$ untyped variants from $400$ haplotypes from Hapmap on chromosome 14. The Non-Parametric Models were trained on chromosome 20 on 1000 Genomes. Only 250 variants were taken per bootstrap.
>
> | Class    | Method     | $k$     | $r^2$    |
> |--------|----|-------|--------|
> | HMM    | Beagle  | 1828     | $0.891 \pm 0.008$   |
> | Non-Parametric Models     | MSA Transformer               | 650       | $0.921 \pm 0.007$ |
> |         | NPSSM  | 1828    | $0.919 \pm 0.007$ |
>
>
> ### Table 9 - Ablations
> Imputation Performance ($r^2$) evaluated on 9159 variants from 516 haplotypes on chromosome 20. Models were trained for 100,000 steps, each with the same model configuration. Flattened is taking the input H_ref and flattening it down into a single sequence (i.e. laying all the sequences in $\text{H}_{\text{ref}}$ back to back).
> | Model                                                                  | $r^2 \pm \sigma$    |
> |------------------------------------------------------------------------|---------------------|
> | SSM_{attr} $\mapsto$ SSM_{data} | $0.9406 \pm 0.0026$ |
> | SSM_{data}  $\mapsto$ SSM_{attr}       | $0.9308 \pm 0.0029$ |
> | Flattened                                                              | $0.8241 \pm 0.0067$ |
> | $\text{SSM}_{\text{attr}}$ Only                                        | $0.8206 \pm 0.0074$ |
> | $\text{SSM}_{\text{data}}$  Only                                       | $0.7783 \pm 0.0049$ |

---

### Official Review · Reviewer_XWVJ · 2024-11-04

**Soundness:** 3
**Presentation:** 3
**Contribution:** 2
**Rating:** 5
**Confidence:** 3

**Summary:**

1. The paper introduces NPSSM, which adapts state-space models (SSMs) into a non-parametric model for  genotype imputation and protein mutation prediction tasks.
2. NPSSM replaces attention with bidirectional Mamba-based SSM layers which achieves a linear complexity in reference set size, an advantage over Transformers which scale quadratically.
3. NPSSM demonstrates competitive performance with transformer-based models

**Strengths:**

1. The paper is well written.
2. NPSSM achieves competitive performance against SOTA transformer model.
3. Memory usage is low due to the use of SSMs, a known benefit that allows for efficient handling of large datasets and contexts.

**Weaknesses:**

**Limited Novelty**: The primary contribution appears to be the replacement of transformers with SSMs, gaining known benefits like linear scalability, handling of long-range dependencies through selectivity. However, these advantages largely derive from established SSM properties rather than novel methodological improvements.

**Ablating the use of Attribute-Specific Components**: The introduced attribute and data-specific SSM layers may benefit from ablation studies. For instance, it is unclear how much of the heavy-lifting is done by the two components and can we remove one of them. Should the order of SSM application be Attribute and then Data or vice versa. Can we instead flatten the sequence.

**Possibly Insufficient Evaluation**: The model is tested on a limited set of baselines. I am not an expert in these tasks but I feel that expanding the number of tasks and baseline models would strengthen the claims of generalizability.

**Questions:**

N/A please see weakness

---

> ### Author Response · Authors · 2024-11-22
> **Response to XWVJ - Novelty and Limited Evaluation**
>
> We thank the reviewer for their feedback
> ### Concern 1:  Novelty
>
> Our novelty lies in the following contributions:
> * We show for the first time that state-space models can be applied to a new task: non-parametric modeling of datapoints and sequence alignments. This expands the scope of tasks that can be addressed with SSMs.
> * We show that SSMs over MSAs are more scalable and performant than transformers over MSAs. The MSATransformer has been immensely important in biological applications (e.g., protein LMs, AlphaFold); improving it has potential for high impact. In that regards, our novelty relative to Mamba is analogous to the novelty of MSATransformer relative to the transformer.
> * We additionally derive a novel probabilistic formulation of these models and a meta-learning objective that goes beyond classical masked training of MSA models. We report that this training is crucial to solve the imputation task (see Table 5, a model trained only on a MLM objective version achieves only $85\\%$ of the performance our full setup achieves ).
> * While our architecture builds on top of BiMamba, producing a performant model requires solving a number of technical challenges. We report in the following sections the effect of these changes.
> * Designing a loss function (Figure 4)
> * Annealing schedules (Table 5)
> * Masking strategies (Table 4, Figure 3)
>
>  Genotype imputation has largely worked off the same model (Li and Stephens HMM) for over 2 decades now, and we show that these models have the potential to supplant these existing models. This however did not come from applying SSMs to the problem, a significant amount of engineering work was required.
>
>
> ### Concern 2: Limited Eval
>  We do the following to address our insufficient evaluation:
>
> 2a. Add Baslines:
>   1. We Add Semi-Parametric Inducing Point Networks **(SPIN) as a baseline**. This model is another non-parametric model, but it achieves linear scaling in the context size and was also evaluated in a Genotype Imputation setting against multiple other linear scaling architectures.
>   2. We Add in **Minimac4 as a baseline**, another HMM based method. This includes all commonly used methods for Imputation.
>
> 2b. **Add Datasets** We add further evaluation (Tables 10 and 11).
>   1.  (**Addition: New Dataset**) Table 11 (results on chr14 on HapMap genomes ) - We add results taking the models trained on a portion of chr20 on 1k genomes and evaluating them on chr14 of HapMap, an entirely different dataset. We split the HapMap dataset into a HapMap specific reference panel and a HapMap specific evaluation set. Given the different chromsome from training, and the context set being used, we argue there is no possiblity for any sort of data leakage and this adequately asses generalization. Note that this dataset is a worst case scenario for long context models (see Questions 3 for details), Yet we still observe HMM methods < NPSSM $\\approx$MSA-Transformer on the set of data.
>   2.  (**Addition: Different Chromosome**) Table 10 (results on chr14 on 1k genomes) - We add results taking the models trained on a portion of chr20 and evaluate them on chr14. We argue that chr14 shows entirely different patterns of relationships between variants and samples (e.g. even if 2 samples/haplotypes/datapoints were full copies of each on chr20, this need not hold for a different chromosome), and so chr14 should be considered as it's own dataset for the purposes of assesing generalizability. We observe HMMs < MSA Transformer < NPSSM on this set of data.
>
> 2c. **Add Statistical Significance**. We re-compute by bootstraping our results (10 bootstrap samples of $\sim 20\\%$ of the data by default). We then report the mean $r^2$ and $\\sigma$ of these trials and generally show HMM Methods < MSA-Transformer < NPSSM. For example, we now see that NPSSM is significantly better than HMMs in all experiments. In addition, we now also show the error rate$(1-r²)$ to better illustrate the performance gains. In Table 2 we see NPSSM is $\sim 12\\%$ better than the best HMM in terms of error rate (0.057 vs 0.050)

---

> ### Author Response · Authors · 2024-11-22
> **Ablations and Relevant Tables**
>
> We attach a copy of the relevant tables below for your convenience.
>
> Concern 3: Ablations
> We incorporate these suggestions as Table 9 (attached below for convenience). Briefly, we trained 5 separate models with the exact configurations (i.e. same number of parameters, same losses, etc) for 100k steps. The only differences for each model were:
>
> * The order of the layers
>   * This has a noticeable effect on the performance of the model, although it is possible that this difference may disappear for longer training budgets.
> * Flattening the input sequence. Instead of having an input of [num_datapoints, num_attributes, embed_dim], we flatten this into [num_datapoints * num_attributes, embed_dim]. The information contained is the same as the base model, but without the MSA structure.
>   * Removing the structure of the sequence severely impacts performance. It may be possible that training for long might result in a decreased performance gap, but at 100k steps there is a significant decrease in performance when removing the structure of the problem.
> * SSM_attr only - We model each relationship between each attribute (variants) in a sample sample.
>   * This is equivalent to ablating away the context set, the model is given a batch of partially masked inputs and is trying to unmask each sequence given only the unmasked variants of each individual sequence.
> * SSM_data only - We model the relationship between all the data points for a given variant.
>   * This model doesn’t learn much, the performance plateaus after ~10k steps. Given that the input sequence to this model is the sequence of a given attribute, it is likely that this version of the model is essentially learning a weighted average of the provided sequences. Given that the performance of this model is very similar to that of the KNN baseline, is is probable that this is the case. Given the results of these ablations, we believe it to be the case that the structure input and both the attribute and datapoint level layers are important to the model.
>
> We hope that if this response adequately addressed the reviewers concerns, they will consider updating their score.
>
> ### Table 9 - Ablations
> Imputation Performance ($r^2$) evaluated on 9159 variants from 516 haplotypes on chromosome 20. Models were trained for 100,000 steps, each with the same model configuration. Flattened is taking the input H_ref and flattening it down into a single sequence (i.e. laying all the sequences in H_ref back to back).
> | Model                                                                  | $r^2 \pm \sigma$    |
> |------------------------------------------------------------------------|---------------------|
> | SSM_{attr} $\mapsto$ SSM_{data} | $0.9406 \pm 0.0026$ |
> | SSM_{data}  $\mapsto$ SSM_{attr}       | $0.9308 \pm 0.0029$ |
> | Flattened          | $0.8241 \pm 0.0067$ |
> | $\text{SSM}_{\text{attr}}$ Only       | $0.8206 \pm 0.0074$ |
> | $\text{SSM}_{\text{data}}$  Only      | $0.7783 \pm 0.0049$ |
>
> ## Table 2
> | Class     | Method             | $k$                   | $r^2$                | $(1-r²)$
> |---------------|------------|-------|----------------------|-----|
> | Traditional ML        | KNN      | 4388       | $0.758 \\pm 0.005$    |  $0.242 \pm 0.005$ |
> |      | LR     | NA     | $0.882 \\pm 0.006$    | $0.118 \pm 0.006$ |
> | HMM    | Beagle      | 4388           | $0.943 \\pm 0.002$    | $0.057 \pm 0.002$ |
> |      | Minimac4 | 4388           | $0.931 \\pm 0.003$    | $0.069 \pm 0.003$ |
> |      | Impute     | 4388            | $0.935 \\pm 0.003$ | $0.065 \pm 0.003$ |
> | Non-Parametric Models | SPIN      | 4388    | $0.868 \\pm 0.010$    | $0.132 \pm 0.010$ |
> |       | MSA Transformer                  | 650     | $0.946 \\pm 0.002$  | $0.054 \pm 0.002$ |
> |      | NPSSM                      | 2000    | $0.950 \\pm 0.002$  | $0.050 \pm 0.002$ |
> ### Table 10 - Chr14 1k Genomes
>
> Imputation performance ($r^2$) evaluated on $19,369$ untyped variants (dev set) from $516$ haplotypes on chromosome 14. The Non-Parametric Models were trained on chromosome 20, and given the nature of the problem there should be no information flow between chromosomes.
> | Class         | Method      | $k$    | $r^2$     |
> |---------|----|------|-------|
> | HMM             | Beagle   | 4388       | $0.964 \pm 0.002$   |
> | Non-Parametric Models        | MSA Transformer      | 650     | $0.963 \pm 0.002$ |
> |         | NPSSM   | 2000      | $0.967 \pm 0.002$ |
>
> ### Table 11 - Chr14 HapMap
> Imputation performance ($r^2$) evaluated on $962$ untyped variants from $400$ haplotypes from Hapmap on chromosome 14. The Non-Parametric Models were trained on chromosome 20 on 1000 Genomes. Only 250 variants were taken per bootstrap.
>
> | Class    | Method     | $k$     | $r^2$    |
> |--------|----|-------|--------|
> | HMM    | Beagle  | 1828     | $0.891 \pm 0.008$   |
> | Non-Parametric Models     | MSA Transformer               | 650       | $0.921 \pm 0.007$ |
> |         | NPSSM  | 1828    | $0.919 \pm 0.007$ |

---

### Official Review · Reviewer_zj3C · 2024-11-04

**Soundness:** 2
**Presentation:** 1
**Contribution:** 2
**Rating:** 3
**Confidence:** 4

**Summary:**

The authors introduce the non-parametric state space model (NPSSM), which scales linearly with the number of data point unlike other attention-based models. The proposed approaches is applied in the context of genotype imputation and meta learning.

**Strengths:**

The proposed approach is simple and inherits the linear scaling of state space models such as BiMamba.

**Weaknesses:**

The methodological component of the paper reads excessively like a pure application (without methodological novelty). First the model basically applies BiMamba twice without further modification. Second, a big  emphasis of the proposed approach is its linear scaling, however, this is not discussed or analyzed in the methods, basically, because it translates directly from BiMamba.

The first experiment (protein analysis) is underwhelming. First they show that increasing k does not improve the performance of neither of the approaches considering, thus defeating the need of a more expressive, but more expensive, model. Second, the model considered had 1M parameters vs. 3.5M in Notin et al. (2023b). Why not consider the larger model at least for NPSSM? Also, from Table 1, it doe not look like PNPT scales (in memory) much worse than NPSSM? One would expect the factor (~2) not to remain that similar for K=1000 and K=1500.

The second experiment is not very convincing because the advantage of NPSSM over MSA Transformer is not clear. This because i) results (Tables 2 and 3) are only presented on a single dataset, ii) the variability and dependency of the main performance characteristics on hyperparameter choices is not clear (though explored via ablation), and iii) computational cost is briefly illustrated in Figure 5. However, is is not clear which context size is used for the main experiments in Tables 2 and 3 and why given the relationship between context size and performance (in Figure 2), the proposed model is not more much better than MSA Transformer in terms of r2.

**Questions:**

From Table 3 and Figure 2 it is not clear how the MSA Transformer can reach a r2=0.956 when it runs out of memory before reaching r2=0.95 in Figure 2?

Why not considering the experiment in Table 6, but starting from the best model in Table 3? Note also that the value of k used in the main experiments is not noted.

---

> ### Author Response · Authors · 2024-11-22
> **Response Part 1**
>
> We thank the reviewer for their feedback
> ### Concern 1:  Novelty
>
> Our novelty lies in the following contributions:
> * We show for the first time that state-space models can be applied to a new task: non-parametric modeling of datapoints and sequence alignments. This expands the scope of tasks that can be addressed with SSMs.
> * We show that SSMs over MSAs are more scalable and performant than transformers over MSAs. The MSATransformer has been immensely important in biological applications (e.g., protein LMs, AlphaFold); improving it has potential for high impact. In that regards, our novelty relative to Mamba is analogous to the novelty of MSATransformer relative to the transformer.
> * We additionally derive a novel probabilistic formulation of these models and a meta-learning objective that goes beyond classical masked training of MSA models. We report that this training is crucial to solve the imputation task (see Table 5, a model trained only on a MLM objective version achieves only $85\\%$ of the performance our full setup achieves ).
> * While our architecture builds on top of BiMamba, producing a performant model requires solving a number of technical challenges. We report in the following sections the effect of these changes.
> * Designing a loss function (Figure 4)
> * Annealing schedules (Table 5)
> * Masking strategies (Table 4, Figure 3)
>
>  Genotype imputation has largely worked off the same model (Li and Stephens HMM) for over 2 decades now, and we show that these models have the potential to supplant these existing models. This however did not come from applying SSMs to the problem, a significant amount of engineering work was required.
>
> We argue that the linear scaling is analyzed, particularly in Figure 2 where we show performance as a function of context set and Figure 5 where we show how the linear scaling of SSMs allows them to outperform Transformer based architectures through the Virtue of having a larger context set in imputation.
>
> ### Concern 2: Imputation Experiments
>  We do the following to address our insufficient evaluation:
>
> 2a. Add Baslines:
>   1. We Add Semi-Parametric Inducing Point Networks **(SPIN) as a baseline**. This model is another non-parametric model, but it achieves linear scaling in the context size and was also evaluated in a Genotype Imputation setting against multiple other linear scaling architectures.
>   2. We Add in **Minimac4 as a baseline**, another HMM based method.
>
> 2b. **Concern: Insufficient Datasets - Add Datasets** We add further evaluation (Tables 10 and 11).
>   1.  (**Addition: New Dataset**) Table 11 (results on chr14 on HapMap genomes ) - We add results taking the models trained on a portion of chr20 on 1k genomes and evaluating them on chr14 of HapMap, an entirely different dataset. We split the HapMap dataset into a HapMap specific reference panel and a HapMap specific evaluation set. Given the different chromsome from training, and the context set being used, we argue there is no possiblity for any sort of data leakage and this adequately asses generalization. Note that this dataset is a worst case scenario for long context models (see Questions 3 for details), Yet we still observe HMM methods < NPSSM $\\approx$MSA-Transformer on the set of data.
>   2.  (**Addition: Different Chromosome**) Table 10 (results on chr14 on 1k genomes) - We add results taking the models trained on a portion of chr20 and evaluate them on chr14. We argue that chr14 shows entirely different patterns of relationships between variants and samples (e.g. even if 2 samples/haplotypes/datapoints were full copies of each on chr20, this need not hold for a different chromosome), and so chr14 should be considered as it's own dataset for the purposes of assesing generalizability. We observe HMMs < MSA Transformer < NPSSM on this set of data.
>
> 2c. **Concern: Unclear Gains - Add Statistical Significance**. We re-compute by bootstraping our results (10 bootstrap samples of $\sim 20\\%$ of the data by default). We then report the mean $r^2$ and $\\sigma$ of these trials and generally show HMM Methods < MSA-Transformer < NPSSM. For example, we now see that NPSSM is significantly better than HMMs in all experiments. In addition, we now also show the error rate$(1-r²)$ to better illustrate the performance gains. In Table 2 we see NPSSM is $\sim 12\\%$ better than the best HMM in terms of error rate (0.057 vs 0.050)
>
> 2d. **Concern: Unclear Hyper parameter Choices - Add Details** - MSA-Transformer and NPSSM use many of the same hyperparameters (e.g. they both use the same annealed loss, the same weighted aux loss, loss schedules, masking strategies, etc). The hyper param choices are always the ones that showed the best performance in section 4.  The largest differences between MSA-Transfromer and NPSSM in terms of hyperparms is the **context set size, which has been added to the relevant tables**.

---

> ### Author Response · Authors · 2024-11-22
> **Response Part 2**
>
> ## Concern 3: PNPT Experiments
> 3a. Performance seems to be invariant to context set size K
> * This is true for some tasks. Certain tasks did see an increase in performance with model size (e.g. TAT_HV1BR_Fernandes_2016 @ k=1000 = 0.577, k=1500 = 0.595, k=2000 = 0.605), but on average most tasks we evaluated did not show an appreciable difference based on context size. However we do show performance matching existing methods on these tasks while consuming less memory, and this was important for the experiments in Imputation.
>
> 3b. Why not bigger models (at least for NPSSM)
> * For the initial experiments, we wanted to keep the number of parameters as close as possible. If we train a larger NPSSM model, we gain modest performance gains in a few tasks, but on average scaling up to 3M parameters does not result in a performance gain.
>
> 3c. Memory Scaling not as expected
> * The 2x memory scaling does not hold for larger $k$. For example k=2000 uses ~29 Gb of Memory.
>
> ## Questions
> 1. From Table 3 and Figure 2 it is not clear how the MSA Transformer can reach a r2=0.956 when it runs out of memory before reaching r2=0.95 in Figure 2?
>     > Figure 2 and Table 3 are on different sets of data, Figure 2 and Table 2 are both on a subset ($1.5\\%$ of chr20, 9k untyped variants) and Table 3 is on the entirety of chr20 (670k untyped variants).
> 2.  Table 2 and Figure 2 are on a subset of Table 3.
> Why not considering the experiment in Table 6, but starting from the best model in Table 3? Note also that the value of k used in the main experiments is not noted.
>     > The results from Table 3 are the best models from Table 6. Specifically, the models in Table 2, Table 3, and Figure 1 are the best performance models from Table 6. There is an Errata for Table 6, the best MSA Transformer model (with k=650) should be 0.9461 (consistent with the 0.946 in Table 2 and Figure 1).
> 3. Why does there seem to be a marginal return on context set size?
> >  Context size is well known to be an important factor in imputation performance, but depending on the dataset used its benefits can be attenuated. For example, if my reference panel contains the mother and father of the query, I should be able to nearly perfectly impute the query with a context size of 4, and more context would yield marginal returns.  However, if instead we have 1000s of samples where eachonly matches the query at $\sim 1\\%$, then the context size is very important since the larger the context size the more likely we can find the relevant information we need to impute the query. For example, the HapMap dataset (Table 11) is known to contain many smaller groups of related individuals, so the benefit of larger context sizes is limited. But even then we still see better performance from the Non-parametric models compared to the HMMs. In practice however, imputation is usually performed with large set of moderately related individuals, where context size really helps.
>
> We hope that if this response adequately addressed the reviewers concerns, they will consider updating their score.
>
> ## Table 2
> | Class                       | Method             | $k$      | $r^2$                | $(1-r²)$
> |----------|------|----------|-------|-----|
> | Traditional ML   | KNN      | 4388       | $0.758 \\pm 0.005$    |  $0.242 \pm 0.005$ |
> |      | LR     | NA   | $0.882 \\pm 0.006$    | $0.118 \pm 0.006$ |
> | HMM    | Beagle    | 4388    | $0.943 \\pm 0.002$    | $0.057 \pm 0.002$ |
> |      | Minimac4 | 4388    | $0.931 \\pm 0.003$    | $0.069 \pm 0.003$ |
> |      | Impute     | 4388     | $0.935 \\pm 0.003$ | $0.065 \pm 0.003$ |
> | Non-Parametric Models | SPIN   | 4388      | $0.868 \\pm 0.010$    | $0.132 \pm 0.010$ |
> |       | MSA Transformer |   650    | $0.946 \\pm 0.002$  | $0.054 \pm 0.002$ |
> |      | NPSSM    | 2000   | $0.950 \\pm 0.002$  | $0.050 \pm 0.002$ |
>
>
> ## New Figures
>
> ### Table 10 - Chr14 1k Genomes
>
> Imputation performance ($r^2$) evaluated on $\sim 19,369$ untyped variants (dev set) from $516$ haplotypes on chromosome 14. The Non-Parametric Models were trained on chromosome 20, and given the nature of the problem there should be no information flow between chromosomes.
> | Class    | Method   | $k$  | $r^2$     |
> |------|----|-------|---------|
> | HMM      | Beagle   | 4388     | $0.964 \pm 0.002$   |
> | Non-Parametric Models   | MSA Transformer      | 650   | $0.963 \pm 0.002$ |
> |                 | NPSSM      | 2000   | $0.967 \pm 0.002$ |
>
> ### Table 11 - Chr14 HapMap
> Imputation performance ($r^2$) evaluated on $962$ untyped variants from $400$ haplotypes from Hapmap on chromosome 14. The Non-Parametric Models were trained on chromosome 20 on 1000 Genomes. Only 250 variants were taken per bootstrap.
>
> | Class       | Method          | $k$         | $r^2$      |
> |------|-------|-------|-----|
> | HMM   | Beagle | 1828     | $0.891 \pm 0.008$   |
> | Non-Parametric Models    | MSA Transformer   | 650     | $0.921 \pm 0.007$ |
> |            | NPSSM        | 1828   | $0.919 \pm 0.007$ |

---

> > ### Comment · Reviewer_zj3C · 2024-11-27
> >
> > Thanks to the authors for their response. However, concerns regarding novelty, performance relative to K and sizing of the NPSSM remain, thus the score will remain unchanged.

---

> > > ### Author Response · Authors · 2024-12-03
> > > **Follow up response**
> > >
> > > We thank the reviewer for engaging with us.
> > > ### Concern: Novelty
> > > We’d like to reiterate that although the application of SSM is a natural continuation of existing work, it is not one that necessarily comes for free and still requires study to verify its applicability. To that end, we study its application to a new and relevant domain in genotype imputation and find that SSMs significantly outperform not only existing HMM based methods but also previous transformer based approaches even when applying the best training strategies and objectives we attempted. We argue that adapting existing ML based methods to this task is non-trivial and showed that with a proper set-up  we can provide an alternative to existing methods that have largely been based off the same HMM for nearly two decades, with the NPSSM specifically being significantly better than the HMM baselines on all datasets we evaluated.
> > >
> > > We believe this work provides a specific and exciting application space where DL models can offer increased performance over existing methods and present the best performing model (NPSSMs) so far, showing the applicability of SSMs to these types of models.
> > > ### Concern: Performance Relative to $k$
> > > We're unsure of the exact ambiguity here, so we'd like to re-iterate:
> > > 1. We have added in the size $k$ used for each model to the relevant tables, examples attached below.
> > > ### Table 2
> > > Performance on Chr20
> > > | Class   | Method | $k$ | $r^2$ |
> > > |---------|------|------|------|
> > > | Traditional ML | KNN | 4388 | $0.758 \\pm 0.005$    |
> > > |   | LR| NA | $0.882 \\pm 0.006$    |
> > > | HMM| Beagle   | 4388 | $0.943 \\pm 0.002$    |
> > > |  | Minimac4  | 4388 | $0.931 \\pm 0.003$    |
> > > | | Impute     | 4388 | $0.935 \\pm 0.003$ |
> > > |Non-Parametric Models | SPIN | 4388 | $0.868 \\pm 0.010$    |
> > > | | MSA Transformer | 650 | $0.946 \\pm 0.002$  |
> > > | | NPSSM | 2000 | **$0.950 \\pm 0.002$**  |
> > >
> > > ### Table 10 - Chr14 1k Genomes
> > >
> > > Imputation performance ($r^2$) evaluated on $19,369$ untyped variants (dev set) from $516$ haplotypes on chromosome 14. The Non-Parametric Models were trained on a different chromosome (chromosome 20).
> > > | Class | Method | $k$ | $r^2$ |
> > > |------|---|--|---|
> > > | HMM | Beagle   | 4388 | $0.964 \\pm 0.002$   |
> > > | Non-Parametric Models  | MSA Transformer| 650 | $0.963 \\pm 0.002$ |
> > > | | NPSSM | 2000 | **$0.967 \\pm 0.002$**|
> > >
> > > 2. We additionally have Table 6 that shows the performance of both MSA-Transformer and NPSSM for various settings of $k$ both at training and during evaluation, clearly showing the positive relationship between the size of $k$ and the performance of the models holding everything else equal. As shown in Table 2 and 10 the differences between 650 (largest MSA $k$) and 2000 (largest NPSSM $k$) are statistically significant (p-val = 0.003 <0.05 using Welch's T-test) .
> > > ### Table 6
> > > Imputation performance ($r^2$) evaluated on 9159 untyped variants from $516$ haplotype on chromosome 20, for various values of $k$ during training and during evaluation.
> > > | Method | $k$ during training  | $k$ during eval | eval $r^2$ |
> > > |----|------|---|----|
> > > | MSA Transformer |  150 | 150| 0.9407|
> > > | MSA Transformer |  150 | 300| 0.9367|
> > > | MSA Transformer |  150 | 650| 0.9156|
> > > | MSA Transformer |  300 | 300| 0.9440|
> > > | MSA Transformer |  300 | 650| 0.9429|
> > > | MSA Transformer |  650 | 650| 0.9461|
> > > | NPSSM| 150 | 150  | 0.9428|
> > > | NPSSM| 150 | 300  | 0.9444|
> > > | NPSSM| 150 | 650  | 0.9445|
> > > | NPSSM| 300 | 300  | 0.9474|
> > > | NPSSM| 300 | 650  | 0.9489|
> > > | NPSSM| 300 | 1000 | 0.9489|
> > > | NPSSM| 650 | 650  | 0.9495|
> > > | NPSSM| 650 | 1000 | 0.9499|
> > > | NPSSM| 650 | 2000 | 0.9501|
> > >
> > > ### Concern: NPSSM Size
> > > We believe the reviewer may be referring to the model configuration and size of the models used. We use 700k parameter models for both MSA Transformer and NPSSM and include the specific model configurations and hyperparameters in Tables 7 and 8 in the Appendix. We include those tables again below for convenience.
> > >
> > > We hope this addresses the reviewers concerns and they consider updating their score.
> > > ### Table 7
> > > Configuration for 700k parameter NPSSM model
> > > | Embedding dim $d$            | 156    |
> > > |--------|-----|
> > > | Number of NPSSM layers       | 2      |
> > > | Initial learning rate   | $1e-3$ |
> > > | $p_{\text{mask}}$     | 0.10   |
> > > | Optimizer  | Adam   |
> > > | effective batch size   | 128    |
> > > | Starting $\\lambda$     | 1.0    |
> > > | $\\lambda$ annealing schedule | linear |
> > > | Ending $\\lambda$ value   | 0.5    |
> > >
> > > ### Table 8
> > > Configuration for 700k MSA Model
> > > | Embedding dim $d$    | 100    |
> > > |----|------|
> > > | Num attention heads    | 10     |
> > > | Number of MSA-Transformer layers | 4      |
> > > | Activation Dropout   | 0.1    |
> > > | Attention Dropout   | 0.1    |
> > > | Initial learning rate    | $6e-4$ |
> > > | $p_{\\text{mask}}$     | 0.10   |
> > > | Optimizer    | Adam   |
> > > | effective batch size  | 128    |
> > > | Starting $\\lambda$  | 1.0    |
> > > | $\\lambda$ annealing schedule    | linear |
> > > | Ending $\\lambda$ value  | 0.5    |
> > > | Datapoint Positional Embedding   | None   |
> > > | Tied Row Attention    | True   |

---

### Author Response · Authors · 2024-11-22
**General Response to Common Concerns**

## Common Feed Back
We thank the reviewers for their helpful feedback. Many reviewers shared a few common points of criticism so we'll address them here in addition to individually responding to every reviwer.
### Concern 1:  Novelty
Multiple reviewers identified the limited novely of this work as a weakness. The claim is that a significant portion of our advantages come for free from applying state space models to this problem (e.g. linear scaling).

Our novelty lies in the following contributions:
* We show for the first time that state-space models can be applied to a new task: non-parametric modeling of datapoints and sequence alignments. This expands the scope of tasks that can be addressed with SSMs.
* We show that SSMs over MSAs are more scalable and performant than transformers over MSAs. The MSATransformer has been immensely important in biological applications (e.g., protein LMs, AlphaFold); improving it has potential for high impact. In that regards, our novelty relative to Mamba is analogous to the novelty of MSATransformer relative to the transformer.
* We additionally derive a novel probabilistic formulation of these models and a meta-learning objective that goes beyond classical masked training of MSA models. We report that this training is crucial to solve the imputation task (see Table 5, a model trained only on a MLM objective version achieves only $85\\%$ of the performance our full setup achieves ).
* While our architecture builds on top of BiMamba, producing a performant model requires solving a number of technical challenges. We report below the effect of these changes.
Designing a loss function (Figure 4)
Annealing schedules (Table 5)
Masking strategies (Table 4, Figure 3)
etc.

 Genotype imputation has largely worked off the same model (Li and Stephens HMM) for over 2 decades now, and we show that these models have the potential to supplant these existing models. This however did not come from applying SSMs to the problem, a significant amount of engineering work was required.

### Concern 2: Insufficient Evaluation
Multiple reviewers identified that our evaluation was insufficient. We agree and do the following:
1. Add Baslines:
    1. We Add Semi-Parametric Inducing Point Networks **(SPIN) as a baseline**. This model is another non-parametric model, but it achieves linear scaling in the context size and was also evaluated in a Genotype Imputation setting against multiple other linear scaling architectures.
    2. We Add in **Minimac4 as a baseline**, another HMM based method.
2. Add Datasets. A common criticism is that we do not have enough varied datasets to accurately assay generalization. We Add Tables 10 and 11 to address this.
    1.  (**Addition: New Dataset**) Table 11 (results on chr14 on HapMap genomes ) - We add results taking the models trained on a portion of chr20 on 1k genomes and evaluating them on chr14 of HapMap, an entirely different dataset. We split the HapMap dataset into a HapMap specific reference panel and a HapMap specific evaluation set. Given the different chromsome from training, and the context set being used, we argue there is no possiblity for any sort of data leakage and this adequately asses generalization. Note that this dataset is a worst case scenario for long context models (see reviewer ZJ3C concern 3 context size). Yet we still observe HMM methods < NPSSM $\approx$MSA-Transformer on the set of data.
    2.  (**Addition: Different Chromosome**) Table 10 (results on chr14 on 1k genomes) - We add results taking the models trained on a portion of chr20 and evaluate them on chr14. We argue that chr14 shows entirely different patterns of relationships between variants and samples (e.g. even if 2 samples/haplotypes/datapoints were full copies of each on chr20, this need not hold for a different chromosome), and so chr14 should be considered as it's own dataset for the purposes of assesing generalizability. We observe HMMs < MSA Transformer < NPSSM on this set of data.
    3. (Clarification) Table 3 - Table 3 (results on full chr20) is included to show the generalizability of Non-parametric methods. NPSSM and MSA-Transformer in Table 3 are trained on the first $\sim 1.5\\%$ of chr20, and then evaled on all of it. While it's the same chr, the $\sim 98.5\\%$ of chr20 that the models were not trained on exhibits different relationships between attributes/variants than was observed during training.
3. **Add Statistical Significance**. We re-compute by bootstraping our results (10 bootstrap samples of $\sim 20\\%$ of the data by default). We then report the mean $r^2$ and $\\sigma$ of these trials and generally show HMM Methods < MSA-Transformer < NPSSM. For example, we now see that NPSSM is significantly better than HMMs in all experiments. In addition, we now also show the error rate$(1-r²)$ to better illustrate the performance gains. In Table 2 we see NPSSM is $\sim 12\\%$ better than the best HMM in terms of error rate (0.057 vs 0.050)

---

### Author Response · Authors · 2024-11-22
**Changelong and Updated Tables**

# Updates
We list the changes we make and the updated Tables for easy reference.
## Changelog
1. Added bootstrapping.
2. Added SPIN baseline (constant attnetion DL model), Minimac4 (another HMM method) and $(1-r²)$ column to Table 2.
3. Added context set sizes (k) .
4. Added Ablations on layer ordering and ablating specific layers of NPSSM (Table 9). All models ablated have the same configuration (loss, number of parameters, etc), the changes are mainly in the structure/order of the input.
5. Added imputation results on chr14 (Table 10). Models were trained on chr20, no further training was done.
6. Added imputation results on HapMAP dataset on chr14 (Table 11) using HapMAP as the reference panel. Models were trained on chr20 on 1k genomes.
7. Errata in Table 6: MSA Transformer train=650, eval=650: 0.9481 -> 0.9461.
8. Additionally in the text we make clearer that Tables 2,3 and Figure 1 all use the models with the best performance from Table 6.
9. Typos and misc formatting fixes.

## Updated Figures

### Table 2 (New Baselines)
| Class                       | Method             | $k$                   | $r^2$                | $(1-r²)$
|------------------------------|--------------------------|-----------------------|----------------------|-----|
| Traditional ML        | KNN                              | 4388                  | $0.758 \\pm 0.005$    |  $0.242 \pm 0.005$ |
|                                | LR                               | NA                    | $0.882 \\pm 0.006$    | $0.118 \pm 0.006$ |
| HMM                  | Beagle      | 4388           | $0.943 \\pm 0.002$    | $0.057 \pm 0.002$ |
|                            | Minimac4 | 4388           | $0.931 \\pm 0.003$    | $0.069 \pm 0.003$ |
|                            | Impute     | 4388            | $0.935 \\pm 0.003$ | $0.065 \pm 0.003$ |
| Non-Parametric Models | SPIN                             | 4388                  | $0.868 \\pm 0.010$    | $0.132 \pm 0.010$ |
|                                        | MSA Transformer                  | 650                   | $0.946 \\pm 0.002$  | $0.054 \pm 0.002$ |
|                                        | NPSSM                      | 2000                  | $0.950 \\pm 0.002$  | $0.050 \pm 0.002$ |


## New Figures
### Table 9 - Ablations
Imputation Performance ($r^2$) evaluated on 9159 variants from 516 haplotypes on chromosome 20. Models were trained for 100,000 steps, each with the same model configuration. Flattened is taking the input H_ref and flattening it down into a single sequence (i.e. laying all the sequences in $\\text{H}_{\\text{ref}}$ back to back).
| Model                                                                  | $r^2 \pm \sigma$    |
|------------------------------------------------------------------------|---------------------|
| SSM_{attr} $\mapsto$ SSM_{data} | $0.9406 \pm 0.0026$ |
| SSM_{data}  $\mapsto$ SSM_{attr}       | $0.9308 \pm 0.0029$ |
| Flattened                                                              | $0.8241 \pm 0.0067$ |
| $\text{SSM}_{\text{attr}}$ Only                                        | $0.8206 \pm 0.0074$ |
| $\text{SSM}_{\text{data}}$  Only                                       | $0.7783 \pm 0.0049$ |
### Table 10 - Chr14 1k Genomes

Imputation performance ($r^2$) evaluated on $\sim 19,369$ untyped variants (dev set) from $516$ haplotypes on chromosome 14. The Non-Parametric Models were trained on chromosome 20, and given the nature of the problem there should be no information flow between chromosomes.
| Class                            | Method                        | $k$                   | $r^2$               |
|----------------------------------|-------------------------------|-----------------------|---------------------|
| HMM             | Beagle   | 4388                  | $0.964 \pm 0.002$   |
| Non-Parametric Models                                  | MSA Transformer               | 650                   | $0.963 \pm 0.002$ |
|                                  | NPSSM                   | 2000                  | $0.967 \pm 0.002$ |

### Table 11 - Chr14 HapMap
Imputation performance ($r^2$) evaluated on $962$ untyped variants from $400$ haplotypes from Hapmap on chromosome 14. The Non-Parametric Models were trained on chromosome 20 on 1000 Genomes. Only 250 variants were taken per bootstrap.

| Class                            | Method                        | $k$                   | $r^2$               |
|----------------------------------|-------------------------------|-----------------------|---------------------|
| HMM             | Beagle   | 1828                  | $0.891 \pm 0.008$   |
| Non-Parametric Models                                  | MSA Transformer               | 650                   | $0.921 \pm 0.007$ |
|                                  | \methodabrv                   | 1828                  | $0.919 \pm 0.007$ |

---

### Meta-Review · Area_Chair_wCr1 · 2024-12-18

**Metareview:**

The paper introduces Non-Parametric State Space Models (NPSSM) which scale linearly with the number of data points, unlike attention-based models that scale quadratically.  This linear scaling is inherited from state space models (SSMs) like BiMamba. NPSSMs show competitive performance against existing non-parametric attention-based models.  They also outperform HMM-based methods in genotype imputation tasks.

However, the reviewers felt that methodological novelty is limited, and that the paper could benefit from a more extensive evaluation. It is tested on a limited set of baselines and datasets. The performance gains of NPSSM over other models, particularly in terms of r2, are not clearly presented and could be further clarified.

The authors did spent a considerable effort during the rebuttal period for improving the paper (added baslelines, ablations) but the paper probably deserves another round of reviews.

For these reasons I recommend rejection but encourage the authors to further improve their manuscript and resubmit it to another conference.

**Additional Comments On Reviewer Discussion:**

- Reviewers pointed out the need for a more thorough evaluation, including additional baselines and datasets.
- Authors responded by adding new baselines, such as SPIN and Minimac4, and incorporating new datasets, including HapMap and chromosome 14 of the 1000 Genomes Project.

- Discussions revolved around clarifying the performance gains of NPSSM over existing models, particularly concerning the r2 metric.
- The authors provided further details on hyperparameter choices and the relationship between context set size and performance.

- Reviewers suggested conducting ablation studies to understand the contribution of different components of the NPSSM model.
The authors performed ablations on layer ordering and the removal of specific layers, demonstrating the importance of both the attribute and data point level layers, as well as the structure of the input.

---

### Decision · Program_Chairs · 2025-01-22

Reject